# PTR: Precision-Driven Tool Recommendation for Large Language Models

## Abstract

By augmenting Large Language Models (LLMs) with external tools, their capacity to solve complex problems has been significantly enhanced. However, despite ongoing advancements in the parsing capabilities of LLMs, incorporating all available tools simultaneously in the prompt remains impractical due to the vast number of external tools. Consequently, it is essential to provide LLMs with a precise set of tools tailored to the specific task, considering both quantity and quality. Current tool retrieval methods primarily focus on refining the ranking list of tools and directly packaging a fixed number of top-ranked tools as the tool set. However, these approaches often fail to equip LLMs with the optimal set of tools prior to execution, since the optimal number of tools for different tasks could be different, resulting in inefficiencies such as redundant or unsuitable tools, which impede immediate access to the most relevant tools. This paper addresses the challenge of recommending precise toolsets for LLMs. We introduce the problem of tool recommendation, define its scope, and propose a novel Precision-driven Tool Recommendation (PTR) approach. PTR captures an initial, concise set of tools by leveraging historical tool bundle usage and dynamically adjusts the tool set by performing tool matching, culminating in a multi-view-based tool addition. Additionally, we present a new dataset, RecTools, and a metric, TRACC, designed to evaluate the effectiveness of tool recommendation for LLMs. We further validate our design choices through comprehensive experiments, demonstrating promising accuracy across two open benchmarks and our RecTools dataset. We release our code and dataset at https://anonymous.4open.science/r/PTR-65DD to support further research in tool recommendation.

## 1 Introduction

Large Language Models (LLMs) have established themselves as powerful intermediaries, demonstrating remarkable impacts across a variety of downstream tasks, including text generation, code debugging, and personalized recommendations (Brown et al., 2020; Touvron et al., 2023; Nam et al., 2024; Chen et al., 2024; Zhao et al., 2024). However, as these models continue to evolve, they still struggle to solve highly complex problems due to limitations arising from their pre-training data (Mialon et al., 2023; Mallen et al., 2022; Yuan et al., 2023). To expand the potential of LLMs in managing more complex tasks efficiently, recommendations at various levels have been increasingly applied to LLMs. Typically, memory recommendations (Borgeaud et al., 2022) and knowledge-based recommendations (Gao et al., 2023; Hu et al., 2023) enhance consistency and context awareness in ongoing tasks for LLMs, while data augmentation recommendations (Xu et al., 2020) facilitate the inclusion of additional data to augment training. Furthermore, architecture recommendations (Elsken et al., 2019; Fedus et al., 2022) and prompt recommendations (Shin et al., 2020; Pryzant et al., 2023; Liu et al., 2023) optimize efficiency and generate more relevant outputs. Simultaneously, to reduce the cognitive load on LLMs and enhance their complex problem-solving capabilities by enabling actions beyond natural language processing, it is crucial to augment LLMs with recommendations of optimal external tool sets, an aspect currently lacking in existing recommendation frameworks for LLMs. Furthermore, this approach will be helpful to address the challenge of input length limitations encountered when incorporating a large number of external tools into the prompt. Providing LLMs with a precise and dynamically adaptable recommended toolset can help to enhance the effectiveness of LLM's task-solving ability.

Considering that the capability of LLMs to master and control external tools is instrumental in overcoming some of their fundamental weaknesses, the field of tool retrieval—which aims to identify the top-K most suitable tools for a given query from a vast set of tools—has been increasingly explored. The advent of tool retrieval (Zhuang et al., 2023; Li et al., 2023; Tang et al.,

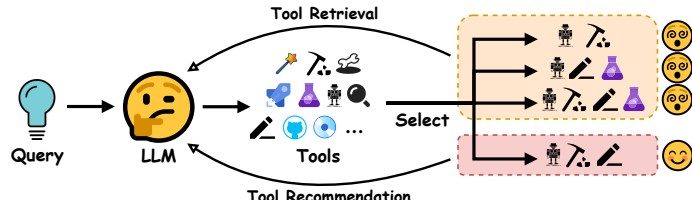

Figure 1: Tool retrieval often provides a broad and variable number of tools with inconsistent quality, whereas tool recommendation delivers a precise, high-quality set of tools directly.

2023; Yang et al., 2024) signifies a nuanced evolution, most directly employing term-based methods (Sparck Jones, 1972; Robertson et al., 2009) or semantic-based techniques (Kong et al., 2023; Yuan et al., 2024; Gao et al., 2024). Generally, the primary objective of these methods is to refine the ranked list of tools and subsequently select a fixed number of tools from the top (top-K) (Qu et al., 2024a; Zheng et al., 2024; Qu et al., 2024b). Although such approaches have demonstrated good performance when retrieving a single tool (Patil et al., 2023; Xu et al., 2023) or a small number of tools (generally fewer than three) (Qin et al., 2023; Huang et al., 2023), they remain susceptible to under-selection or over-selection, as illustrated in Figure.1. This limitation may prevent LLMs from addressing the current query or cause them to over-interpret the query, thereby reducing the effectiveness of LLMs in solving complex problems with external tools. Additionally, the validation of these methods often relies on datasets that use a fixed number of tools for each query, meaning that during testing, the number of tools to be used is known in advance—an unrealistic scenario in practical applications where the number of tools needed can vary dynamically. Therefore, recommending a precise and dynamically adjustable set of external tools to LLMs in a single step prior to query execution is increasingly important. This approach not only enhances the thoroughness of problem-solving but also improves efficiency by reducing the need to execute additional tools.

To address these limitations, we first provide a comprehensive explanation of tool recommendation and clearly define the problem, considering the lack of definition and the incompleteness of goals pursued by existing tool retrieval methods. Toward this objective, we propose PTR, a novel model-agnostic **P**recision-Driven **T**ool **R**ecommendation approach aimed at recommending a precise tool set for LLMs prior to query execution. By leveraging historical tool bundle usage data to uncover patterns of idiomatic use and dependencies between tools, this method is structured into three main stages: *Tool Bundle Acquisition, Functional Coverage Mapping*, and *Multi-view-based Re-ranking*. Initially, using traditional pre-trained language models, we acquire semantic matching information between queries and previously used tool bundles, thereby addressing potential performance issues of these models in zero-shot scenarios for tool recommendation tasks. Subsequently, to evaluate the effectiveness of the selected tool bundle in solving the query, LLMs are prompted to match tools with the specific subproblems they can address and to identify unresolved issues. Based on this, a multi-view-based re-ranking method is employed to select tools that can help resolve the identified issues and complement the existing tool sets. More specifically, to address the unresolved issues, we construct the final ranked list by aggregating three tool lists and ranking each tool based on their frequency of occurrence. The ranked tool list, constructed from multiple views, reduces the randomness associated with selecting tools from the entire available set.

Additionally, we construct a dataset, **RecTools**, tailored to specific queries with recommended tool sets. In contrast to previous tool datasets that standardize the number of tools used for each query (Huang et al., 2023) or employ a small number of tools (Qu et al., 2024a), our tool recommendation set incorporates varying numbers of tools for different queries, with up to ten tools used for a single query. This is achieved through an automated process in which LLMs are prompted to generate specific queries to be addressed by given tool bundles. These queries and tool bundles are subsequently evaluated by prompting LLMs to determine whether the selected tools adequately address the corresponding queries, ensuring that neither excess nor insufficient tools are utilized. Dedicated validation and deduplication steps are implemented to ensure the precision of tool usage, thereby enhancing the quality of the tool recommendation set.

Furthermore, traditional retrieval metrics such as Recall (Zhu, 2004) and Normalized Discounted Cumulative Gain (NDCG) (Järvelin & Kekäläinen, 2002), fail to capture the level of precision re-

quired for effective tool recommendation. The absence of necessary tools can lead to the failure of LLMs in performing tasks, while the redundancy of tools may cause LLMs to generate unnecessary responses. This indicates that metrics focusing solely on completeness are inadequate for evaluating tool recommendation tasks. To bridge this gap, we introduce **TRACC**, a novel metric designed to assess tool recommendation performance, considering both the accuracy of the quantity and the quality of the recommended tools. TRACC serves as a reliable indicator of the effectiveness of tool recommendation processes.

To summarize, the main contributions of this work are as follows:

- We introduce tool recommendation as a novel problem, necessitating the provision of precise tool sets to LLMs for a given query. We propose PTR, an effective tool recommendation approach that leverages historical tool bundle information between queries and tools, resulting in a more accurate and comprehensive final recommended tool list.

- We present a new dataset, RecTools, and an effective evaluation metric, TRACC, specifically designed to assess tool recommendation for LLMs. This not only addresses gaps in existing tool sets but also advances future research related to tool recommendation.

- Extensive experiments validate the effectiveness of RecTools and demonstrate the efficacy of PTR in recommending tools for LLMs. The recommended tool sets are both comprehensive and accurate, enhancing the overall performance of LLMs in processing tasks.

## 2 TOOL RECOMMENDATION

Tool retrieval, as discussed in previous, involves generating a comprehensive list of tools that are potentially relevant to a user's query. This approach emphasizes breadth, aiming to maximize the inclusion of pertinent tools. While effective in ensuring extensive coverage, tool retrieval often prioritizes recall over precision, resulting in the inclusion of extraneous tools that may not be essential for the task at hand. Addressing this limitation, we propose a new optimization direction–Tool Recommendation–for LLMs. It aims to ensure that the recommended set of tools aligns closely with the ground-truth set of tools for a task, both in quantity and quality. Specifically, given a user query with a ground-truth toolset $(A, B, C)$, tool recommendation aims to identify precisely $(A, B, C)$, avoiding omissions or the inclusion of redundant tools. Here is the definition of the tool recommendation task:

**Definition 1** *Tool Recommendation: Given a comprehensive set of tools $T = \{T_1, T_2, \ldots, T_n\}$ and a query Q, let $T_{ground} \subseteq T$ denote the ground truth toolset that fully satisfies Q. The objective is to recommend a toolset $T_{recommend} = \{T_1, T_2, \ldots, T_k\}$ from T such that $T_{recommend} = T_{ground}$ and the cardinality constraint $|T_{recommend}| = |T_{ground}|$ holds.*

As discussed in previous, achieving precision in tool recommendation is pivotal for enhancing the performance and reliability of LLMs. By minimizing the inclusion of irrelevant tools, LLMs can reduce computational overhead, streamline task execution, and improve the overall quality of responses. Addressing precised tool recommendation not only mitigates the drawbacks associated with broad tool retrieval but also paves the way for more sophisticated and user-centric LLM applications. This advancement is essential for deploying LLMs in environments where efficiency, accuracy, and user satisfaction are crucial.

## 3 THE PRECISION-DRIVEN TOOL RECOMMENDATION

We introduce a novel approach, Precision-driven Tool Recommendation (PTR), to address the challenges faced by prior research through a three-stage recommendation process: (1) Tool Bundle Acquisition, which involves establishing a potentially useful tool bundle by leveraging past usage patterns across all tool combinations, as opposed to relying solely on instructions for individual tool usage; (2) Functional Coverage Mapping, which entails effectively mapping the tools from the acquired tool bundle to the functionalities of the original query, thereby identifying which tools should be retained and which should be discarded, resulting in any remaining unsolved sub-problems; and

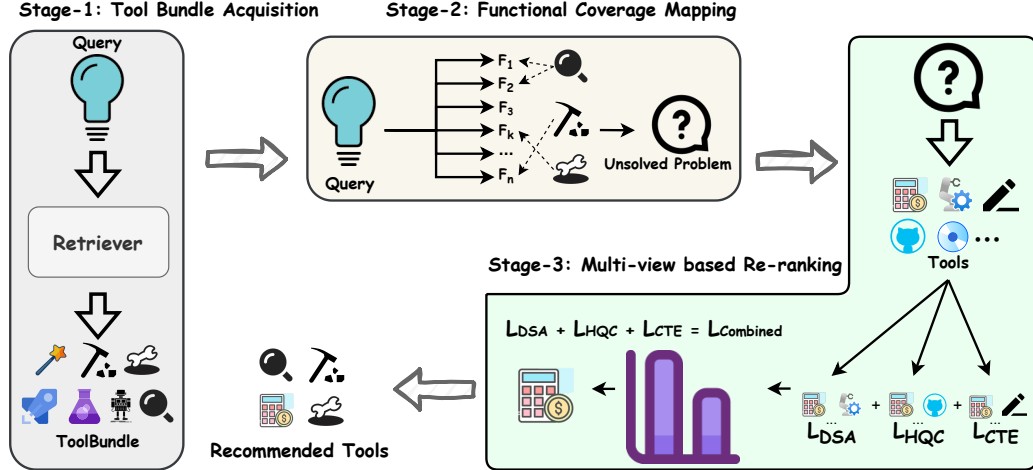

Figure 2: Architecture of the three-stage recommendation framework PTR for tool recommendation.

(3) Multi-view-based Re-ranking, which involves the effective re-ranking of relevant tools from a large tool set, tailored to each unsolved sub-problem identified in the second stage, and selecting the top-ranked tool after re-ranking to complete the final recommended toolset. The overview of our approach is illustrated in Figure.2. Please note that all symbols are globally defined in sections 2 and 3. In the following sections, we present the details of these three PTR recommendation stages.

## 3.1 TOOL BUNDLE ACQUISITION

To obtain a initiate set of tools, we employ an retriever to capture the relevance between historical tool combinations and the current query. Unlike existing methods that focus on retrieving single tools by analyzing the relationship between a query and individual tools, our approach introduces tool bundle retrieval. By leveraging historical tool combinations, we capture a richer contextual relationship between queries and sets of tools that have been used together effectively in the past. This facilitates a more holistic understanding of tool dependencies and synergies, thereby enhancing the relevance of retrieved tool sets for complex queries. Specifically, Let $T = \{T_1, T_2, \ldots, T_n\}$ be the set of all available tools. Let $D = \{(Q_i, B_i)\}_{i=1}^{M}$ represent a set of past queries and their associated tool bundles, where $Q_i$ is a past query, and $B_i$ is the corresponding tool bundle used for $Q_i$, with $B_i \subseteq T$. The collection of unique tool bundles is $B = \{B_1, B_2, \ldots, B_N\}$. Given a new query $Q$, we select a tool bundle $B_K = \{T_1, \ldots, T_z\}$ from $B$ that is most relevant to $Q$ through the retriever, which ideally contains tools potentially useful. The subsequent recommendations operate on this obtained tool bundle—either based on sparse representations or dense representations.

## 3.2 FUNCTIONAL COVERAGE MAPPING

As illustrated in Figure.3, functional coverage mapping presents a structured approach to evaluate and optimize a set of tools in relation to a specific query. By systematically aligning required functionalities with the capabilities of available tools, this method ensures that the toolset comprehensively addresses the user's needs while minimizing redundancies and identifying any gaps, as each tool may correspond to multiple functionalities. At its core, Functional Coverage Mapping aims to determine whether an initial set of tools $B_K = \{T_1, T_2, \ldots, T_z\}$ adequately fulfills a query $Q$ with its key functionalities $F = \{F_1, F_2, \ldots, F_m\}$. Specifically, Functional Coverage Mapping achieves

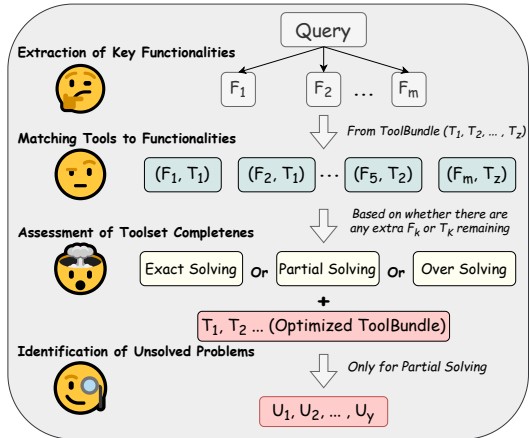

Figure 3: The four stages of Functional Coverage Mapping in PTR.

this objective through four steps: *Extraction of Key Requirements*, *Matching Tools to Functionalities*, *Assessment of Toolset Completeness*, and *Identification of Unsolved Problems*, which are described as follows:

***Extraction of Key Functionalities***. The first step involves decomposing the user's query $Q$ into a set of discrete and actionable functionalities $R$. This extraction ensures a comprehensive understanding of the query that the toolset must address. This extraction is achieved by prompting the language model to identify and enumerate these functionalities directly from the query, ensuring that both explicit and implicit functionalities are captured.

***Matching Tools to Functionalities***. Once the key functionalities $F$ are established, the subsequent phase entails mapping each functionality $F_i$ to the tools $T_j$ within the obtained tool bundle $B_K$. This mapping process determines which tools are capable of fulfilling specific functionalities. To achieve this, targeted prompts are employed with the language model, directing it to associate each functionality with the most suitable tool based on tool descriptions.

***Assessment of Toolset Completeness***. With the mapping $M(F, B_K)$ established, the method evaluates whether the toolset $B_K$ fully addresses all functionalities $F$. This assessment categorizes the toolset into one of three scenarios: (1) Exact Solving: All functionalities are met by all tools without any redundancies; (2) Oversolving: The toolset includes tools that provide functionalities not required by the query; and (3) Partial Solving: Some functionalities remain unfulfilled and some tools remain unused. Based on the identified scenario, the tool bundle is optimized by retaining essential tools and discarding redundant ones. Tools that do not contribute to fulfilling any requirement are removed to streamline the toolset.

***Identification of Unsolved Problems***. In cases of partial solving, the method identifies the remaining unsolved problems directly from the original query $Q$. These unsolved problems $U = \{U_1, U_2, \ldots, U_y\}$ are presented in a format that can be directly utilized in the subsequent recommendation stage. To achieve this, the language model is prompted to extract the unmet functionalities without further functional decomposition. This approach ensures that each unsolved problems retains the context of the original query $Q$, thereby facilitating seamless integration with the following re-ranking method. Furthermore, this direct identification allows for straightforward utilization in the following re-ranking process, where each unsolved problem can be addressed individually.

### 3.3 MULTI-VIEW BASED RE-RANKING

Addressing the challenge of selecting pertinent tools from an extensive toolset to resolve unresolved problems requires comprehensive consideration. The proposed PTR employs a multifaceted similarity evaluation strategy that integrates three essential dimensions of the unresolved problem $U_j$: (1) **Direct Semantic Alignment**, wherein the system quantifies the semantic similarity between the user query and each available tool, ensuring the immediate identification of tools intrinsically aligned with the query's intent; (2) **Historical Query Correlation**, which involves analyzing past queries that closely resemble the current one to extract tools previously utilized in similar contexts, thereby enriching the current toolset with empirically effective solutions while maintaining uniqueness through aggregation and deduplication; and (3) **Contextual Tool Expansion**, which leverages the most relevant tool identified through direct semantic alignment to retrieve additional tools exhibiting high similarity to this primary tool, thereby uncovering supplementary options that may offer complementary or alternative functionalities beneficial to the user's query. The multi-view matching process involves obtaining the tool list $L$ through direct semantic alignment (DSA), historical query correlation (HQC), and contextual tool expansion (CTE), respectively. These three tool lists are then aggregated and ranked according to their frequency of occurrence, with the most frequent tools being selected. After performing the multi-view-based re-ranking for each unsolved problem, the top-ranked tool in each list is selected and added to the final recommended toolset. In some cases, it is also possible that this tool already exists in the toolset acquired from the second-stage recommendation; in such instances, the tool will be ignored. The algorithm for multi-view-based re-ranking is summarized in Algorithm.1.

## 4 DATASETS AND METRICS

---

**Algorithm 1** Multi-view Based Re-ranking

---

**Require:** Unresolved problem $U_j$, Toolset $T = \{T_1, T_2, \ldots, T_n\}$, Historical queries $Q = \{Q_1, Q_2, \ldots, Q_m\}$, $\text{Select}_K$ represents the function that selects the top $K$ candidates with the highest similarity, $\sigma$ indicates the similarity measure.

**Ensure:** Recommended Tool $\mathcal{T}$.

1: Initialize lists: $L_{\text{DSA}}$, $L_{\text{HQC}}$, $L_{\text{CTE}}$.
     //Direct Semantic Alignment
2: $L_{\text{DSA}} \leftarrow \text{Select}_K (\{T_i \in T \mid \sigma(U_j, T_i)\})$     ▷ Directly obtain the tools most relevant to the given query
     //Historical Query Correlation
3: $L_{\text{HistoricalQuery}} \leftarrow \text{Select}_K (\{Q_i \in Q \mid \sigma(U_j, Q_i)\})$     ▷ Retrieve the most relevant past queries.
4: **for** each query $Q_i$ in $L_{\text{HistoricalQuery}}$ **do**
5:     **for** each tool $T_l$ used in $Q_i$ **do**
6:         Add $T_l$ to $L_{\text{HQC}}$
7:     **end for**
8: **end for**
9: Remove duplicates from $L_{\text{HQC}}$.
     //Contextual Tool Expansion
10: **if** $L_{\text{DSA}}$ is not empty **then**
11:     $T_{\text{primary}} \leftarrow L_{\text{DSA}}[0]$     ▷ Obtain the most relevant tool identified in the first stage.
12:     $L_{\text{CTE}} \leftarrow \text{Select}_K (\{T_i \in T \mid \sigma(T_{\text{primary}}, T_i)\})$
13: **end if**
14: Combine lists: $L_{\text{Combined}} \leftarrow L_{\text{DSA}} + L_{\text{HQC}} + L_{\text{CTE}}$
15: Count frequency of each tool in $L_{\text{Combined}}$.
16: Rank tools by frequency in descending order.
17: Select the top ranked tool as $\mathcal{T}$.
      **return** $\mathcal{T}$.

---

*Datasets.* To verify the effectiveness of PTR, we utilize three datasets for tool recommendation: ToolLens (Qu et al., 2024a), MetaTool (Huang et al., 2023), and a newly constructed dataset, **RecTools**. We randomly select 20% of each dataset to serve as the test data. Both Tool-

Table 1: Statistics of the experimental datasets.

| Feature | ToolLens | MetaTool | RecTools |
|---|---|---|---|
| **Tools per Query** | 1-3 | 2 | 1-10 |
| **Unified used tool number** | ✓ | ✗ | ✓ |
| **Exact Solving Test** | 6.34% | 55.1% | 61.3% |

Lens and MetaTool focus on multi-tool tasks, leading us to select them as the primary datasets for our experiments. While ToolLens uniquely emphasizes creating queries that are natural, concise, and intentionally multifaceted, MetaTool is a benchmark designed to evaluate whether LLMs possess tool usage awareness and can correctly choose appropriate tools. However, both datasets impose a low upper limit on the number of tools used per query. As the capabilities of LLMs continue to develop, more tools need to be recommended to solve increasingly complex problems, thereby limiting the applicability of these datasets. Additionally, all queries in these two datasets utilize a fixed number of tools, which not only fails to fully simulate the dynamic nature of tool usage in real-world scenarios but also introduces bias in the subsequent testing of the method. Most importantly, since tool recommendation focuses on the precision of the recommended toolset, the test datasets require that each query be exactly solvable by the provided tools (Exact Solving). Using one fewer tool leads to partial solving, while using one additional tool results in oversolving. To validate the effectiveness of the two datasets, we first employ GPT-4o as an evaluator to determine whether the provided toolset can achieve an "Exact Solving" outcome for each query. Subsequently, for each query, we randomly remove one tool from the corresponding toolset and prompt GPT-4o to assess whether the modified toolset can achieve a "Partial Solving" outcome. Queries and their respective toolsets that meet the criteria for both evaluations are considered qualified. The performance of these two datasets is not ideal. Based on these limitations, we constructed a new dataset, **RecTools**, where queries do not have a uniform number of tools and have a high upper limit on the number of tools used (details in Appendix.A). Additionally, RecTools significantly outperforms ToolLens and Metatool in the GPT-4o "Exact Solving" test. The statistics of the three datasets are summarized in Table.1. Specifically, for all (query, tools) pairs involving the use of two and three tools, the success rates of RecTools reached 76% and 89%, respectively.

*Metrics.* As evaluation metrics for tool recommendation, following previous work focusing on tool retrieval (Gao et al., 2024; Qu et al., 2024b), the widely used retrieval metrics are Recall and NDCG. However, they do not adequately address the requirements for accuracy in both the number of rec-

Table 2: Performance comparisons of PTR under different methods within different backbones on ToolLens, MetaTool, and RecTools datasets. "N/A" indicates that this method works alone. The best results are bolded, the best results of each colunmn are denoted as "*".

| Methods | Framework | ToolLens | | | MetaTool | | | RecTools | | |
|---|---|---|---|---|---|---|---|---|---|---|
| | | Recall@K | NDCG@K | TRACC | Recall@K | NDCG@K | TRACC | Recall@K | NDCG@K | TRACC |
| Random | N/A | 0.036 | 0.061 | 0.034 | 0.133 | 0.202 | 0.133 | 0.137 | 0.271 | 0.097 |
| | +PTR+open-mistral-7b | 0.185 | 0.225 | 0.145 | 0.608 | 0.785 | 0.505 | 0.457 | 0.756 | 0.235 |
| | +PTR+GPT-3.5-turbo | 0.213 | 0.282 | 0.172 | 0.645 | 0.823 | 0.543 | 0.475 | 0.784 | 0.288 |
| | +PTR+GPT-4o | **0.227** | **0.303** | **0.187** | **0.663** | **0.843** | **0.562** | **0.492** | **0.802** | **0.305** |
| BM25 | N/A | 0.131 | 0.194 | 0.125 | 0.429 | 0.603 | 0.429 | 0.486 | 0.596 | 0.382 |
| | +PTR+open-mistral-7b | 0.206 | 0.254 | 0.162 | 0.659 | 0.834 | 0.554 | 0.524 | 0.795 | 0.355 |
| | +PTR+GPT-3.5-turbo | 0.247 | 0.313 | 0.193 | 0.694 | 0.874 | 0.593 | 0.541 | 0.815 | 0.408 |
| | +PTR+GPT-4o | **0.261** | **0.331** | **0.208** | **0.712** | **0.892** | **0.612** | **0.545** | **0.810** | **0.414** |
| Contriever | N/A | 0.130 | 0.190 | 0.121 | 0.439 | 0.672 | 0.439 | 0.367 | 0.786 | 0.304 |
| | +PTR+open-mistral-7b | 0.208 | 0.256 | 0.164 | 0.662 | 0.837 | 0.557 | 0.512 | 0.773 | 0.342 |
| | +PTR+GPT-3.5-turbo | 0.250 | 0.316 | 0.196 | 0.697 | 0.877 | 0.596 | 0.528 | 0.792 | 0.396 |
| | +PTR+GPT-4o | **0.264** | **0.334** | **0.211** | **0.715** | **0.895** | **0.615** | **0.559** | **0.834** | **0.426** |
| SBERT | N/A | 0.251 | 0.349 | 0.209 | 0.495 | 0.725 | 0.495 | 0.496 | 0.772 | 0.434 |
| | +PTR+open-mistral-7b | 0.272 | 0.362 | 0.226 | 0.682 | 0.862 | 0.582 | 0.538 | 0.821 | 0.452 |
| | +PTR+GPT-3.5-turbo | 0.308 | 0.403 | 0.252 | 0.723 | 0.902 | 0.623 | 0.555 | 0.840 | 0.484 |
| | +PTR+GPT-4o | **0.322** | **0.422** | **0.268** | **0.741** | **0.921** | **0.642** | **0.572** | **0.859** | **0.501** |
| TAS-B | N/A | 0.279 | 0.381 | 0.263 | 0.657 | 0.897 | 0.657 | 0.509 | 0.841 | 0.454 |
| | +PTR+open-mistral-7b | 0.298 | 0.398 | 0.278 | 0.702 | 0.882 | 0.602 | 0.552 | 0.854 | 0.472 |
| | +PTR+GPT-3.5-turbo | 0.335 | 0.438 | 0.305 | 0.741 | 0.922 | 0.642 | 0.567 | 0.872 | 0.505 |
| | +PTR+GPT-4o | **0.352** | **0.456** | **0.321** | **0.759** | **0.941** | **0.661** | **0.583** | **0.890** | **0.522** |
| SimCSE | N/A | 0.293 | 0.386 | 0.279 | 0.675 | 0.849 | 0.675 | 0.563 | 0.808 | 0.523 |
| | +PTR+opem-mistral-7b | 0.312 | 0.407 | 0.291 | 0.716 | 0.897 | 0.631 | 0.578 | 0.861 | 0.542 |
| | +PTR+GPT-3.5-turbo | 0.350 | 0.448 | 0.319 | 0.756 | 0.937 | 0.671 | 0.594 | 0.879 | 0.575 |
| | +PTR+GPT-4o | **0.368***| **0.467***| **0.336***| **0.774***| **0.956***| **0.690***| **0.609***| **0.896***| **0.591*** |

ommended tools and the specific tools recommended, disregarding the impact of differences in size between the tool sets. Therefore, to further tailor the assessment to the challenges of tool recommendation tasks, we introduce a new metric, named **TRACC**. This metric is designed to measure the extent to which the recommended toolset aligns with the ground-truth set in terms of both the accuracy of the number of tools and the accuracy of the tools themselves:

$$\text{TRACC} = \left(1 - \frac{1}{|A \cup B|} \cdot |n_2 - n_1|\right) \cdot ACC$$

where $A$ denotes the ground-truth tool set and $B$ represents the recommended tool set. The cardinalities of $A$ and $B$ are denoted by $n_1$ and $n_2$, respectively. And $|A \cup B|$ signifies the cardinality of the union of $A$ and $B$. ACC represents $\frac{|A \cap B|}{n_1}$, where $|A \cap B|$ indicates the size of the their intersection.

## 5 EXPERIMENTS

### 5.1 IMPLEMENTATION DETAILS

*Baselines.* We considered the following baselines: **Random**, which randomly select from historical tools; **BM25** (Robertson et al., 2009), a classical sparse retrieval method that extends TF-IDF by leveraging term frequency and inverse document frequency of keywords; **Contriever** (Izacard et al., 2021), which utilizes inverse cloze tasks, cropping for positive pair generation, and momentum contrastive training to develop dense retrievers; **SBERT** (Reimers & Gurevych, 2019), a library providing BERT-based sentence embeddings. Specifically, we use all-mpnet-base-v2; **TAS-B** (Hofstätter et al., 2021), the retriever introduces an efficient topic-aware query and balanced margin sampling technique; And **SimCSE** (Gao et al., 2021), a simple contrastive learning framework that greatly advances state-of-the-art sentence embeddings.

Besides, we initially implement the PTR using the open source model open-mistral-7b, due to its cost-effectiveness. Subsequently, we evaluate PTR with the model GPT-3.5-turbo and GPT-4o, to determine its effectiveness when employing a more advanced model. For evaluation metrics, in addition to the specifically designed TRACC metric, we also calculate Recall@K and NDCG@K, reporting these metrics with K set to the size of the ground-truth tool set.

### 5.2 EXPERIMENTAL RESULTS

Table 2 presents the main results of the PTR applied to ToolLens, MetaTool, and RecTools using various models and unsupervised retrievers. Based on these findings, we draw the following observations and conclusions.

Table 3: Ablation study of Tool Bundle Acquisition (w/o Tool Bundle Acquisition).

| Methods (w/o Tool Bundle Acquisition) | ToolLens | | | MetaTool | | | RecTools | | |
|---|---|---|---|---|---|---|---|---|---|
| | Recall@K | NDCG@K | TRACC | Recall@K | NDCG@K | TRACC | Recall@K | NDCG@K | TRACC |
| PTR + open-mistral-7b | 0.221 | 0.264 | 0.171 | 0.695 | 0.882 | 0.612 | 0.532 | 0.912 | 0.270 |
| PTR + GPT-3.5-turbo | 0.264 | 0.381 | 0.208 | 0.724 | 0.919 | 0.656 | 0.541 | 0.913 | 0.430 |
| PTR + GPT-4o | **0.283** | **0.391** | **0.235** | **0.745** | **0.922** | **0.677** | **0.581** | **0.916** | **0.439** |

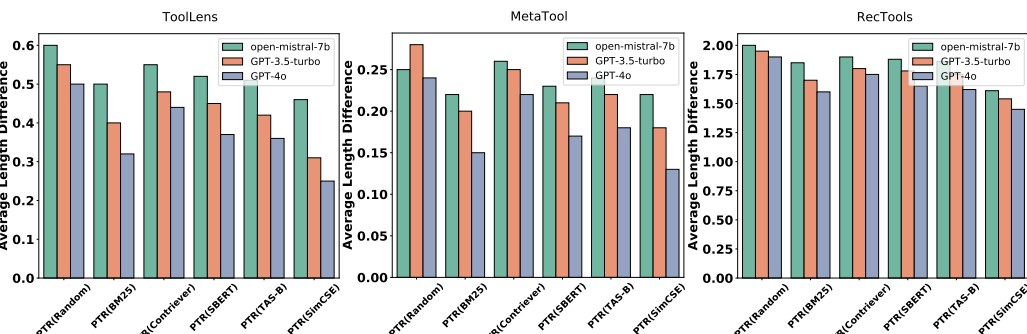

Figure 4: The average length difference between the recommended tool set and the ground truth tool set for each method and backbone.

We first observe that the MetaTool dataset yields notable performance, whereas other datasets exhibit comparatively standard. This discrepancy can be attributed to the presence of relatively straightforward patterns within the MetaTool dataset, which motivates us the construction of a structurally diversified and high-quality tool-query dataset. Furthermore, the Random baseline indicates that random sampling of tool bundles leads to relatively poor performance, whereas other unsupervised retrievers outperform the Random baseline, particularly in the ToolLens dataset. This suggests that, although the latter two phases of the PTR can supplement or refine the recommended tool set, employing a targeted bundle in the early stages can enhance PTR performance. Conversely, the Sim-CSE approach demonstrated a significant improvement over the Random baseline, especially when utilizing GPT-4o as the backbone. Absolute Recall@K improvements of 0.141, 0.111, and 0.117 were observed on the ToolLens, MetaTool, and RecTools datasets, respectively, highlighting the SimCSE method's capability to leverage tool bundle information for more effective tool recommendation. Despite this advantage, all the methods fall short in the TRACC metric, which is specifically designed for evaluating precision in tool recommendation. This suggests that, although effective for tool retrieval tasks, Recall@K and NDCG@K may not fully satisfy the unique requirements of tool recommendation. Additionally, the results demonstrate that PTR consistently achieves strong performance when utilizing GPT-4o, confirming that PTR remains beneficial for tool recommendation even when employing more capable backbone models.

Overall, PTR exhibits effectiveness across all metrics and datasets, attributable to its implementation of a three-stage recommendation framework. This framework comprises tool bundle acquisition, functional coverage mapping for the deletion or retention of tools, and multi-view-based re-ranking for the addition of tools. By employing this structured approach, PTR dynamically addresses the entirety of the query, thereby facilitating the recommendation of a precise and well-tailored tool set.

## 5.3 FURTHER ANALYSIS

In this section, we conduct an in-depth analysis of the effectiveness for PTR, using the same datasets and evaluation metrics. The results are presented in Table 3.

***w/o Tool Bundle Acquisition***. This variant omits the tool bundle acquisition stage, resulting in queries being exclusively mapped to unresolved problems without any existing recommended tools. The observed decline in performance for this variant further supports the effectiveness of tool bundles in identifying potential recommended tools, thereby refining the unresolved problems and achieving precise tool recommendations. Moreover, as illustrated in Table 3, the random approach alone is largely ineffective for tool recommendations. However, as presented in Table 2, when combined with functional coverage mapping and multi-view-based re-ranking, the final recommendation performance improves significantly. This underscores the importance of the last two recommendation stages.

***Performance w.r.t to accuracy in quantity.*** Furthermore, to evaluate the performance of PTR in terms of tool number precision, we calculate the average length difference between the recommended tool set and the ground truth tool set for each method and backbone. Figure.4 demonstrates the effectiveness of PTR in maintaining consistency in the number of tools. In the MetaTool and ToolLens dataset, which exhibits relatively simple and small patterns, PTR clearly shows its effectiveness. Regarding our RecTools dataset, which has a variable structure and involves a wide range of tools for each query, the average length difference is effectively controlled within a considerable range, especially when it comes to the Embedding method.

## 6 RELATED WORK

### 6.1 RECOMMENDATION FOR LLMS

Recent research has explored a variety of recommendation techniques to enhance Large Language Models (LLMs), integrating capabilities across multiple dimensions. Data recommendation (Xu et al., 2020; Ouyang et al., 2022) is crucial for selecting relevant datasets to fine-tune models for specific domains, ensuring ongoing performance improvements. Memory recommendation (Borgeaud et al., 2022; Gao & Zhang, 2024a) facilitates the retrieval of relevant past experiences or interactions, improving continuity, consistency, and long-term context in multi-turn conversations. Knowledge base recommendation (Gao et al., 2023; Hu et al., 2023; Petroni et al., 2019; Lewis et al., 2020) enhances factual grounding by retrieving the most pertinent information from external sources, ensuring that model outputs are accurate and up to date. Architecture recommendation (Elsken et al., 2019; Fedus et al., 2022) optimizes model performance by dynamically selecting the most appropriate model components or layers to activate for different tasks, thereby improving efficiency. Lastly, prompt recommendation (Shin et al., 2020; Reynolds & McDonell, 2021; Li & Liang, 2021; Wang et al., 2022; Liu et al., 2023) guides LLMs in utilizing the most effective input prompts, thereby enhancing the quality of generated responses through optimized input-output interactions. Together, these recommendation techniques form a comprehensive framework that enhances the adaptability, efficiency, and task-specific performance of LLMs. However, there remains a lack of research on tool recommendation. In this work, we motivate to seek to provide a clear definition of tool recommendation and proposes an effective recommendation method. Additionally, new datasets and metrics are created to advance research in this area.

### 6.2 TOOL RETRIEVAL

Initially, term-based methods such as BM25 (Robertson et al., 2009) and TF-IDF (Sparck Jones, 1972) were employed to measure the similarity between queries and tool documents by identifying exact term matches. Subsequently, with the development of dense retrievers (Karpukhin et al., 2020; Guu et al., 2020; Xiong et al., 2020), the semantic relationships between queries and tool descriptions have been more effectively captured through neural networks. Recently, novel approaches for training retrievers have emerged. For example, Confucius (Gao et al., 2024) selects tools by defining three levels of scenarios, ranging from easy to difficult, to train and deepen the LLM's understanding of tools. Additionally, execution feedback is iteratively utilized to refine the tool selection process (Wang et al., 2023; Xu et al., 2024). Furthermore, ToolkenGPT (Hao et al., 2024) enhances tool selection by representing each tool as a token ("toolken") and learning an embedding for it, thereby enabling tool calls in the same manner as generating regular word tokens. Moreover, some research has focused on addressing the diversity of retrieval (Carbonell & Goldstein, 1998; Gao & Zhang, 2024b), which can effectively enhance the quality of multiple tools used by query. Despite their comprehensive nature, tool retrieval systems present notable limitations. The inclusion of superfluous tools can introduce noise, thereby interfering with the LLM's performance and task execution, and these systems are often unable to dynamically adjust the toolset. In this work, we extend our approach beyond getting a rough toolset by ensuring that the tools in the recommended toolset are as accurate as possible in terms of both quality and quantity.

## 7 CONCLUSIONS

This study presents a novel challenge, tool recommendation, and offers a precise formalization of the problem. In response, we propose a new approach, PTR, designed to improve the accuracy of tool

recommendations, considering both the quantity and the selection of tools. PTR operates through three key stages: tool bundle acquisition, functional coverage mapping, and multi-view-based reranking. By dynamically adjusting the tool bundle obtained in the first stage—through the addition or removal of tools—PTR progressively refines the recommended toolset. Extensive experiments and detailed analyses showcase PTR's effectiveness in addressing diverse query structures requiring multiple tool recommendations. Furthermore, we introduce RecTools, a new dataset, along with TRACC, a comprehensive evaluation metric. Both serve as valuable contributions to the future research in the field of tool recommendation.

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

# APPENDIX

# A DETAILS OF RECTOOLS

## A.1 DATASET CONSTRUCTION

To construct our dataset, we utilized tools from the MetaTool (Huang et al., 2023) dataset, along with their corresponding descriptions. Since their objective of tools was to address the issue of overlapping—where a single query could be resolved by multiple tools—MetaTool consolidates groups of tools with similar functionalities into a single tool entity. Besides, those tools and their description come from OpenAI's plugin list, making them more practical. In our dataset RecTools, there are 10 usage scenarios in total (from 1-10), where the usage scenarios mean the quantitative classification, like two tools be used together, ten tools be used together. Each scenario of tools usage contains 100 examples. In each scenario, there are 20 different tool combinations. In terms of each combination, we randomly select from all possible combinations(i.e., $\binom{1}{n}, \binom{2}{n}, ..., \binom{10}{n}$). And for each tool combinations, we generate 5 queries. The prompt is as follows:

```
You are an assistant tasked with generating user queries that can be
exclusively solved by a specific set of tools.

**Requirements for the query:**
1. The query must **only** require the functionalities of the selected
tools.
2. All tools in the selected set must be **necessary** to solve the query
.
3. The query should **not** require any tools outside the selected set.
4. The query should be **clear, specific, and realistic**.
5. **Each query should address a different scenario or aspect** to ensure
 uniqueness. Avoid merely rephrasing similar ideas; focus on varied use
cases.

**Selected Tools:**
XX, XXX

**Tool Descriptions:**
- **XX**: Search for podcasts and summarize their content.
- **XXX**: Discover and support restaurants, shops & services near you.

Generate one unique query that meets the above requirements.
```

A.2  DATASET EVALUATION

To ensure precision in tool recommendation, it is crucial that the query is addressed entirely by
the provided tools. If any tool is missing, the query cannot be fully solved, and if an unnecessary
tool is included, the solution becomes redundant or repetitive. We employ GPT-4 as an evalua-
tor to determine whether the provided toolset can achieve an "Exact Solving" outcome for each
query. Subsequently, for each query, we randomly remove one tool from the corresponding toolset
and prompt GPT-4 to assess whether the modified toolset can achieve a "Partial Solving" outcome.
Queries and their respective toolsets that meet the criteria for both evaluations are considered qual-
ified. For the first evaluation, if it achieves "Exact Solving", we give it a score 1, else 0; For the
second evaluation, if it achieves "Partial Solving", we give it a score 1, else 0; For the final score, if
both of them are 1, then 1; else, 0. The prompt is as follows:

```
Prompt1(Before deletion)
**Query:** "XXX"

**Tools:**
- **XX**: xxxxxx
- **XX**: xxxxxx
- **XX**: xxxxxx

**Classification:** (.Categorize the solving scenario into one of the
following:
1. **Exact Solving:** All functionalities are met by all tools without
any redundancies.
2. **Oversolving:** The toolset includes tools that provide
functionalities not required by the query.
3. **Partial Solving:** Some functionalities remain unfulfilled and some
tools remain unused.)

------------------------------------------------

Prompt2(After deletion)
**Query:** "XXX"

**Tools after removing one tool:**
- **XX**: xxxxxx
- **XX**: xxxxxx
```

```
**Classification:** (.Categorize the solving scenario into one of the
following:
1. **Exact Solving:** All functionalities are met by all tools without
any redundancies.
2. **Oversolving:** The toolset includes tools that provide
functionalities not required by the query.
3. **Partial Solving:** Some functionalities remain unfulfilled and some
tools remain unused.)
```

The final output of evaluation is like this:

```
    {
      "query": "XXX",
      "tools_used": [
        "XX",
        "XX"
      ],
      "first_evaluation": "xxx",
      "second_evaluation_after_deletion": "xxx",
      "score": X
    },
```

Listing 1: An full example for evaluation

```
Few-Shot Examples:

**Query:** "I need the latest weather forecast for New York and a
reminder to carry an umbrella."

**Tools:**
- **WeatherTool**: Provide you with the latest weather information.
- **ReminderTool**: No description available.

**Classification:** Exact Solving

**Query:** "Show me the top-rated restaurants nearby and provide a route
to get there."

**Tools:**
- **RestaurantFinder**: No description available.
- **RoutePlanner**: No description available.

**Classification:** Exact Solving

**Query:** "Find me a good book to read and suggest a nearby coffee shop
."

**Tools:**
- **BookRecommender**: No description available.
- **WeatherTool**: Provide you with the latest weather information.

**Classification:** Partial Solving

**Query:** "Provide the current exchange rates and set a reminder to
check them later."

**Tools:**
- **FinanceTool**: Stay informed with the latest financial updates, real-
time insights, and analysis on a wide range of options, stocks,
cryptocurrencies, and more.
- **ReminderTool**: No description available.
- **NewsTool**: Stay connected to global events with our up-to-date news
around the world.

**Classification:** Oversolving
```

**Query:** "I want to track my fitness goals and get news updates."

**Tools:**
- **FitnessTracker**: No description available.
- **NewsTool**: Stay connected to global events with our up-to-date news around the world.

**Classification:** Exact Solving

**Query:** "Schedule a meeting and find the latest sports news."

**Tools:**
- **CalendarTool**: No description available.
- **NewsTool**: Stay connected to global events with our up-to-date news around the world.
- **FinanceTool**: Stay informed with the latest financial updates, real-time insights, and analysis on a wide range of options, stocks, cryptocurrencies, and more.

**Classification:** Oversolving

**Query:** "Research and select appropriate investment options for setting up a trust fund, ensure compliance with relevant laws, and find suitable gifts for beneficiaries to commemorate the establishment of the trust."

**Tools:**
- **FinanceTool**: Stay informed with the latest financial updates, real-time insights, and analysis on a wide range of options, stocks, cryptocurrencies, and more.
- **LawTool**: Enables quick search functionality for relevant laws.
- **GiftTool**: Provide suggestions for gift selection.

**Classification:** (Respond with only one of the following exact phrases : "Exact Solving", "Oversolving", or "Partial Solving". Do not include any additional text or explanations.)

First Evaluation: Exact Solving

Few-Shot Examples:

**Query:** "I need the latest weather forecast for New York and a reminder to carry an umbrella."

**Tools:**
- **WeatherTool**: Provide you with the latest weather information.
- **ReminderTool**: No description available.

**Classification:** Exact Solving

**Query:** "Show me the top-rated restaurants nearby and provide a route to get there."

**Tools:**
- **RestaurantFinder**: No description available.
- **RoutePlanner**: No description available.

**Classification:** Exact Solving

**Query:** "Find me a good book to read and suggest a nearby coffee shop ."

```
**Tools:**
- **BookRecommender**: No description available.
- **WeatherTool**: Provide you with the latest weather information.

**Classification:** Partial Solving

**Query:** "Provide the current exchange rates and set a reminder to
check them later."

**Tools:**
- **FinanceTool**: Stay informed with the latest financial updates, real-
time insights, and analysis on a wide range of options, stocks,
cryptocurrencies, and more.
- **ReminderTool**: No description available.
- **NewsTool**: Stay connected to global events with our up-to-date news
around the world.

**Classification:** Oversolving

**Query:** "I want to track my fitness goals and get news updates."

**Tools:**
- **FitnessTracker**: No description available.
- **NewsTool**: Stay connected to global events with our up-to-date news
around the world.

**Classification:** Exact Solving

**Query:** "Schedule a meeting and find the latest sports news."

**Tools:**
- **CalendarTool**: No description available.
- **NewsTool**: Stay connected to global events with our up-to-date news
around the world.
- **FinanceTool**: Stay informed with the latest financial updates, real-
time insights, and analysis on a wide range of options, stocks,
cryptocurrencies, and more.

**Classification:** Oversolving

**Query:** "Research and select appropriate investment options for
setting up a trust fund, ensure compliance with relevant laws, and find
suitable gifts for beneficiaries to commemorate the establishment of the
trust."

**Tools after removing one tool:**
- **FinanceTool**: Stay informed with the latest financial updates, real-
time insights, and analysis on a wide range of options, stocks,
cryptocurrencies, and more.
- **LawTool**: Enables quick search functionality for relevant laws.

**Classification:** (Respond with only one of the following exact phrases
: "Exact Solving", "Oversolving", or "Partial Solving". Do not include
any additional text or explanations.)

Second Evaluation (After Deletion): Partial Solving
Score for this query: 1

****************************
****************************

{
```

```
        "query": "Research and select appropriate investment options for
        setting up a trust fund, ensure compliance with relevant laws, and
        find suitable gifts for beneficiaries to commemorate the
        establishment of the trust.",
        "tools_used": [
          "FinanceTool",
          "LawTool",
          "GiftTool"
        ],
        "first_evaluation": "Exact Solving",
        "second_evaluation_after_deletion": "Partial Solving",
        "score": 1
}
```

## B  FUNCTIONAL COVERAGE MAPPING

### B.1  EXTRACTION OF KEY FUNCTIONALITIES

```
You are an assistant helping to extract key requirements from user
queries.

Example 1:
User Query:
"I want a website where users can create accounts, post messages, and
follow other users."

Key Requirements:
- Users can create accounts
- Users can post messages
- Users can follow other users

Example 2:
User Query:
"I need an e-commerce platform that supports product listings, shopping
cart functionality, payment processing, and order tracking."

Key Requirements:
- Supports product listings
- Provides shopping cart functionality
- Handles payment processing
- Offers order tracking

Now, given the following user query, extract the key requirements.

User Query:
XXX

Key Requirements:
```

### B.2  MATCHING TOOLS TO FUNCTIONALITIES

```
You are an assistant helping to match tools to requirements, as long as
the tool description can roughly provid the needed information for
requirments, it does not need to be very specific,ignore the proper nouns
.

Available Tools: XX:xxxxx; XX:xxxxxx.

Example 1:
Requirement:
"I want to know the latest news about Tesla"
```

```
Matched Tools:
- NewsTool: Stay connected to global events with our up-to-date news
around the world.

Example 2:
Requirement:
"Please provide me with the current stock price of Apple"

Matched Tools:
- FinanceTool: Stay informed with the latest financial updates, real-time
 insights, and analysis on a wide range of options, stocks,
cryptocurrencies, and more.

Now, for the following requirement, list the tools from the available
tools that can fulfill it.

Requirement:
XXX
XXX
XXX

Matched Tools:
```

## B.3 EXAMPLES

### Listing 2: An example in ToolLens

```
You are an assistant helping to extract key requirements from user
queries.

Example 1:
User Query:
"I want a website where users can create accounts, post messages, and
follow other users."

Key Requirements:
- Users can create accounts
- Users can post messages
- Users can follow other users

Example 2:
User Query:
"I need an e-commerce platform that supports product listings, shopping
cart functionality, payment processing, and order tracking."

Key Requirements:
- Supports product listings
- Provides shopping cart functionality
- Handles payment processing
- Offers order tracking

Now, given the following user query, extract the key requirements.

User Query:
"I'm preparing for a marathon in Paris, France."
---------------------
Key Requirements:
- Marathon preparation
- Location: Paris, France

***************************
***************************
```

```
1026   You are an assistant helping to match tools to requirements, as long as
1027   the tool description can roughly provid the needed information for
1028   requirments, it does not need to be very specific,ignore the proper nouns
1029   .
1030
1031   Available Tools:
       - **Countries**: This gets geo data on a country. Use ISO2 for
1032   country_code.
1033   - **Skyscanner_v2**: Search for a place to get the **entityId** needed in
1034    searching the hotel API.
1035   - **TimeTable Lookup**: Returns the nearest airports for a given latitude
        and longitude
1036
1037   Example 1:
1038   Requirement:
1039   "I want to know the latest news about Tesla"
1040
1041   Matched Tools:
       - NewsTool: Stay connected to global events with our up-to-date news
1042   around the world.
1043
1044   Example 2:
1045   Requirement:
       "Please provide me with the current stock price of Apple"
1046
1047   Matched Tools:
1048   - FinanceTool: Stay informed with the latest financial updates, real-time
1049    insights, and analysis on a wide range of options, stocks,
1050   cryptocurrencies, and more.
1051
1052   Now, for the following requirement, list the tools from the available
       tools that can fulfill it.
1053
1054   Requirement:
1055   "Marathon preparation"
1056
1057   Matched Tools:
1058
1059   You are an assistant helping to match tools to requirements, as long as
1060   the tool description can roughly provid the needed information for
1061   requirments, it does not need to be very specific,ignore the proper nouns
1062   .
1063   Available Tools:
1064   - **Countries**: This gets geo data on a country. Use ISO2 for
1065   country_code.
       - **Skyscanner_v2**: Search for a place to get the **entityId** needed in
1066    searching the hotel API.
1067   - **TimeTable Lookup**: Returns the nearest airports for a given latitude
1068    and longitude
1069
1070   Example 1:
1071   Requirement:
       "I want to know the latest news about Tesla"
1072
1073   Matched Tools:
1074   - NewsTool: Stay connected to global events with our up-to-date news
1075   around the world.
1076
1077   Example 2:
       Requirement:
1078   "Please provide me with the current stock price of Apple"
1079
       Matched Tools:
```

```
1080    - FinanceTool: Stay informed with the latest financial updates, real-time
1081      insights, and analysis on a wide range of options, stocks,
1082    cryptocurrencies, and more.
1083
1084    Now, for the following requirement, list the tools from the available
1085    tools that can fulfill it.
1086
1087    Requirement:
        "Location: Paris, France"
1088
1089    Matched Tools:
1090
1091    Tool Matches:
        - Requirement: 'Marathon preparation' matched with Tools: None
1092    - Requirement: 'Location: Paris, France' matched with Tools: None
1093
1094    Does the toolset exactly solve the query? No
1095    Tools to Keep:
1096
1097    Unsolved Problems:
        - Marathon preparation
1098    - Location: Paris, France
1099
1100
1101                          Listing 3: An example in MetaTool
1102    You are an assistant helping to extract key requirements from user
1103    queries.
1104
1105    Example 1:
        User Query:
1106    "I want a website where users can create accounts, post messages, and
1107    follow other users."
1108
1109    Key Requirements:
        - Users can create accounts
1110    - Users can post messages
1111    - Users can follow other users
1112
1113    Example 2:
1114    User Query:
        "I need an e-commerce platform that supports product listings, shopping
1115    cart functionality, payment processing, and order tracking."
1116
1117    Key Requirements:
1118    - Supports product listings
        - Provides shopping cart functionality
1119    - Handles payment processing
1120    - Offers order tracking
1121
1122    Now, given the following user query, extract the key requirements.
1123
1124    User Query:
        "I'm looking for a family-friendly destination in Europe with good
1125    weather. Can you suggest some options and what the weather will be like
1126    during summer?"
1127    --------------------
1128    Key Requirements Extracted:
1129    - Family-friendly destination in Europe
        - Options about Europe
1130    - Information on weather during summer
1131
1132    ***************************
1133    ***************************
```

```
You are an assistant helping to match tools to requirements, as long as
the tool description can roughly provid the needed information for
requirments, it does not need to be very specific,ignore the proper nouns
.

Available Tools:
- **ResearchFinder**: Tool for searching academic papers.
- **WeatherTool**: Provide you with the latest weather information.

Example 1:
Requirement:
"I want to know the latest news about Tesla"

Matched Tools:
- NewsTool: Stay connected to global events with our up-to-date news
around the world.

Example 2:
Requirement:
"Please provide me with the current stock price of Apple"

Matched Tools:
- FinanceTool: Stay informed with the latest financial updates, real-time
 insights, and analysis on a wide range of options, stocks,
cryptocurrencies, and more.

Now, for the following requirement, list the tools from the available
tools that can fulfill it.

Requirement:
"Family-friendly destination in Europe"

Matched Tools:

You are an AI assistant helping to match tools to requirements, as long
as the tool description can roughly provid the needed information for
requirments, it does not need to be very specific,ignore the proper nouns
.

Available Tools:
- **ResearchFinder**: Tool for searching academic papers.
- **WeatherTool**: Provide you with the latest weather information.

Example 1:
Requirement:
"I want to know the latest news about Tesla"

Matched Tools:
- NewsTool: Stay connected to global events with our up-to-date news
around the world.

Example 2:
Requirement:
"Please provide me with the current stock price of Apple"

Matched Tools:
- FinanceTool: Stay informed with the latest financial updates, real-time
 insights, and analysis on a wide range of options, stocks,
cryptocurrencies, and more.

Now, for the following requirement, list the tools from the available
tools that can fulfill it.

Requirement:
```

```
"Options about Europe"

Matched Tools:

You are an AI assistant helping to match tools to requirements, as long
as the tool description can roughly provid the needed information for
requirments, it does not need to be very specific,ignore the proper nouns
.

Available Tools:
- **ResearchFinder**: Tool for searching academic papers.
- **WeatherTool**: Provide you with the latest weather information.

Example 1:
Requirement:
"I want to know the latest news about Tesla"

Matched Tools:
- NewsTool: Stay connected to global events with our up-to-date news
around the world.

Example 2:
Requirement:
"Please provide me with the current stock price of Apple"

Matched Tools:
- FinanceTool: Stay informed with the latest financial updates, real-time
 insights, and analysis on a wide range of options, stocks,
cryptocurrencies, and more.

Now, for the following requirement, list the tools from the available
tools that can fulfill it.

Requirement:
"Information on weather during summer"

Matched Tools:
WeatherTool: Provide you with the latest weather information.

Tool Matches:
- Requirement: 'Family-friendly destination in Europe' matched with Tools
: None
- Requirement: 'Good weather' matched with Tools: None
- Requirement: 'Information on weather during summer' matched with Tools:
 WeatherTool

Does the toolset exactly solve the query? No
Tools to Keep:
WeatherTool

Unsolved Problems:
- Family-friendly destination in Europe
- Options about Europe
- Information on weather during summer
```

Listing 4: An example in RecTools

```
You are an assistant helping to extract key requirements from user
queries.

Example 1:
User Query:
"I want a website where users can create accounts, post messages, and
follow other users."
```

```
Key Requirements:
- Users can create accounts
- Users can post messages
- Users can follow other users

Example 2:
User Query:
"I need an e-commerce platform that supports product listings, shopping
cart functionality, payment processing, and order tracking."

Key Requirements:
- Supports product listings
- Provides shopping cart functionality
- Handles payment processing
- Offers order tracking

Now, given the following user query, extract the key requirements.

User Query:
"I want to find a local restaurant with a menu that fits my diet plan,
book a table, get astrology insights on the best date for my dinner, and
select a thoughtful gift for my dining companion."
---------------------
Key Requirements Extracted:
- Find a local restaurant
- Provide a menu that fits the user's diet plan
- Book a table
- Offer astrology insights on the best date for dinner
- Select a thoughtful gift for the dining companion

****************************
****************************

You are an assistant helping to match tools to requirements, as long as
the tool description can roughly provid the needed information for
requirments, it does not need to be very specific,ignore the proper nouns
.

Available Tools:
- **DietTool**: A tool that simplifies calorie counting, tracks diet, and
 provides insights from many restaurants and grocery stores. Explore
recipe , menus, and cooking tips from millions of users, and access
recipe consultations and ingredient delivery services from thousands of
stores.
- **GiftTool**: Provide suggestions for gift selection.
- **HousePurchasingTool**: Tool that provide all sorts of information
about house purchasing
- **HouseRentingTool**: Tool that provide all sorts of information about
house renting
- **MemoryTool**: A learning application with spaced repetition
functionality that allows users to create flashcards and review them.
- **RestaurantBookingTool**: Tool for booking restaurant
- **ResumeTool**: Quickly create resumes and receive feedback on your
resume.
- **StrologyTool**: Povides strology services for you.
- **local**: Discover and support restaurants, shops & services near you.

Example 1:
Requirement:
"I want to know the latest news about Tesla"

Matched Tools:
- NewsTool: Stay connected to global events with our up-to-date news
around the world.
```

```
Example 2:
Requirement:
"Please provide me with the current stock price of Apple"

Matched Tools:
- FinanceTool: Stay informed with the latest financial updates, real-time
 insights, and analysis on a wide range of options, stocks,
cryptocurrencies, and more.

Now, for the following requirement, list the tools from the available
tools that can fulfill it.

Requirement:
"Find a local restaurant"

Matched Tools:

You are an assistant helping to match tools to requirements, as long as
the tool description can roughly provid the needed information for
requirments, it does not need to be very specific,ignore the proper nouns
.

Available Tools:
- **DietTool**: A tool that simplifies calorie counting, tracks diet, and
 provides insights from many restaurants and grocery stores. Explore
recipe , menus, and cooking tips from millions of users, and access
recipe consultations and ingredient delivery services from thousands of
stores.
- **GiftTool**: Provide suggestions for gift selection.
- **HousePurchasingTool**: Tool that provide all sorts of information
about house purchasing
- **HouseRentingTool**: Tool that provide all sorts of information about
house renting
- **MemoryTool**: A learning application with spaced repetition
functionality that allows users to create flashcards and review them.
- **RestaurantBookingTool**: Tool for booking restaurant
- **ResumeTool**: Quickly create resumes and receive feedback on your
resume.
- **StrologyTool**: Povides strology services for you.
- **local**: Discover and support restaurants, shops & services near you.

Example 1:
Requirement:
"I want to know the latest news about Tesla"

Matched Tools:
- NewsTool: Stay connected to global events with our up-to-date news
around the world.

Example 2:
Requirement:
"Please provide me with the current stock price of Apple"

Matched Tools:
- FinanceTool: Stay informed with the latest financial updates, real-time
 insights, and analysis on a wide range of options, stocks,
cryptocurrencies, and more.

Now, for the following requirement, list the tools from the available
tools that can fulfill it.

Requirement:
"Provide a menu that fits the user's diet plan"
```

```
Matched Tools:
DietTool: A tool that simplifies calorie counting, tracks diet, and
provides insights from many restaurants and grocery stores. Explore
recipe , menus, and cooking tips from millions of users, and access
recipe consultations and ingredient delivery services from thousands of
stores.

You are an assistant helping to match tools to requirements, as long as
the tool description can roughly provid the needed information for
requirments, it does not need to be very specific,ignore the proper nouns
.

Available Tools:
- **DietTool**: A tool that simplifies calorie counting, tracks diet, and
 provides insights from many restaurants and grocery stores. Explore
recipe , menus, and cooking tips from millions of users, and access
recipe consultations and ingredient delivery services from thousands of
stores.
- **GiftTool**: Provide suggestions for gift selection.
- **HousePurchasingTool**: Tool that provide all sorts of information
about house purchasing
- **HouseRentingTool**: Tool that provide all sorts of information about
house renting
- **MemoryTool**: A learning application with spaced repetition
functionality that allows users to create flashcards and review them.
- **RestaurantBookingTool**: Tool for booking restaurant
- **ResumeTool**: Quickly create resumes and receive feedback on your
resume.
- **StrologyTool**: Povides strology services for you.
- **local**: Discover and support restaurants, shops & services near you.

Example 1:
Requirement:
"I want to know the latest news about Tesla"

Matched Tools:
- NewsTool: Stay connected to global events with our up-to-date news
around the world.

Example 2:
Requirement:
"Please provide me with the current stock price of Apple"

Matched Tools:
- FinanceTool: Stay informed with the latest financial updates, real-time
 insights, and analysis on a wide range of options, stocks,
cryptocurrencies, and more.

Now, for the following requirement, list the tools from the available
tools that can fulfill it.

Requirement:
"Book a table"

Matched Tools:

You are an AI assistant helping to match tools to requirements, as long
as the tool description can roughly provid the needed information for
requirments, it does not need to be very specific,ignore the proper nouns
.

Available Tools:
- **DietTool**: A tool that simplifies calorie counting, tracks diet, and
 provides insights from many restaurants and grocery stores. Explore
```

```
recipe , menus, and cooking tips from millions of users, and access
recipe consultations and ingredient delivery services from thousands of
stores.
- **GiftTool**: Provide suggestions for gift selection.
- **HousePurchasingTool**: Tool that provide all sorts of information
about house purchasing
- **HouseRentingTool**: Tool that provide all sorts of information about
house renting
- **MemoryTool**: A learning application with spaced repetition
functionality that allows users to create flashcards and review them.
- **RestaurantBookingTool**: Tool for booking restaurant
- **ResumeTool**: Quickly create resumes and receive feedback on your
resume.
- **StrologyTool**: Povides strology services for you.
- **local**: Discover and support restaurants, shops & services near you.

Example 1:
Requirement:
"I want to know the latest news about Tesla"

Matched Tools:
- NewsTool: Stay connected to global events with our up-to-date news
around the world.

Example 2:
Requirement:
"Please provide me with the current stock price of Apple"

Matched Tools:
- FinanceTool: Stay informed with the latest financial updates, real-time
 insights, and analysis on a wide range of options, stocks,
cryptocurrencies, and more.

Now, for the following requirement, list the tools from the available
tools that can fulfill it.

Requirement:
"Offer astrology insights on the best date for dinner"

Matched Tools:
StrologyTool: Povides strology services for you.

You are an AI assistant helping to match tools to requirements, as long
as the tool description can roughly provid the needed information for
requirments, it does not need to be very specific,ignore the proper nouns
.

Available Tools:
- **DietTool**: A tool that simplifies calorie counting, tracks diet, and
 provides insights from many restaurants and grocery stores. Explore
recipe , menus, and cooking tips from millions of users, and access
recipe consultations and ingredient delivery services from thousands of
stores.
- **GiftTool**: Provide suggestions for gift selection.
- **HousePurchasingTool**: Tool that provide all sorts of information
about house purchasing
- **HouseRentingTool**: Tool that provide all sorts of information about
house renting
- **MemoryTool**: A learning application with spaced repetition
functionality that allows users to create flashcards and review them.
- **RestaurantBookingTool**: Tool for booking restaurant
- **ResumeTool**: Quickly create resumes and receive feedback on your
resume.
- **StrologyTool**: Povides strology services for you.
- **local**: Discover and support restaurants, shops & services near you.
```

```
Example 1:
Requirement:
"I want to know the latest news about Tesla"

Matched Tools:
- NewsTool: Stay connected to global events with our up-to-date news
around the world.

Example 2:
Requirement:
"Please provide me with the current stock price of Apple"

Matched Tools:
- FinanceTool: Stay informed with the latest financial updates, real-time
  insights, and analysis on a wide range of options, stocks,
cryptocurrencies, and more.

Now, for the following requirement, list the tools from the available
tools that can fulfill it.

Requirement:
"Select a thoughtful gift for the dining companion"

Matched Tools:
GiftTool: Provide suggestions for gift selection.

Tool Matches:
- Requirement: 'Find a local restaurant' matched with Tools: None
- Requirement: 'Provide a menu that fits the user's diet plan' matched
with Tools: DietTool
- Requirement: 'Book a table' matched with Tools: None
- Requirement: 'Offer astrology insights on the best date for dinner'
matched with Tools: StrologyTool
- Requirement: 'Select a thoughtful gift for the dining companion'
matched with Tools: GiftTool

Does the toolset exactly solve the query? No
Tools to Keep: DietTool, StrologyTool, GiftTool

Unsolved Problems:
- Find a local restaurant
- Book a table
```

