# OpenReview forum: "PTR: Precision-Driven Tool Recommendation for Large Language Models"
_ICLR.cc/2025/Conference — ICLR 2025 Conference Withdrawn Submission_

### Official Review · Reviewer_Q3z2 · 2024-10-26

**Soundness:** 1
**Presentation:** 2
**Contribution:** 2
**Rating:** 3
**Confidence:** 4

**Summary:**

This manuscript introduces Precision-driven Tool Recommendation (PTR), a novel approach for recommending precise toolsets to Large Language Models (LLMs). The authors argue that existing tool retrieval methods often suffer from imprecision issue. PTR aims to address this by recommending a toolset that precisely matches the groundtruth set. The manuscript also introduces a new dataset, RecTools, and a new evaluation metric, TRACC, designed specifically for the tool recommendation task.

**Strengths:**

The introduction of the RecTools dataset is a valuable contribution to the research community, providing a resource specifically designed for evaluating tool retrieval and utilization in LLMs.

**Weaknesses:**

1. The manuscript lacks a compelling discussion of the research motivation. While providing LLMs with a concise set of tools intuitively benefits performance in tool-using scenarios, the actual impact of redundancy on performance degradation remains unclear and unexplored. The impact of tool redundancy on LLM performance is not adequately demonstrated, weakening the justification for precision-driven tool recommendation.
2. The experimental evaluation focuses on performance comparison with baselines that don't incorporate the proposed PTR method. However, the manuscript fails to consider the efficiency aspect. The proposed method involves multiple LLM calls, which likely introduce significant computational overhead. The manuscript must discuss the trade-off between performance gains and computational cost.

**Questions:**

1. In line 313, the manuscript mentions a second-turn evaluation where a tool is randomly removed to assess "Partial Solving." This is confusing, especially if the first evaluation already identifies "Oversolving." Could the authors clarify the purpose and logic behind this two-step evaluation process and provide a clearer explanation?

2. In line 350, the manuscript introduces the TRACC metric, which incorporates the absolute difference in the number of tools between the recommended set and the ground truth set ($|n_2 - n_1|$). Could the authors further elaborate on the rationale behind this specific component? Why are traditional metrics like the F1-score, which considers both precision and recall, insufficient for this task?

3. Table 2 demonstrates the performance variations arising from using different LLMs. An analysis of the efficiency and cost associated with each LLM choice and without LLM would enhance the evaluation.

---

### Official Review · Reviewer_7pBz · 2024-10-31

**Soundness:** 2
**Presentation:** 3
**Contribution:** 2
**Rating:** 5
**Confidence:** 3

**Summary:**

This paper introduces a novel problem called "tool recommendation" for Large Language Models (LLMs), which aims to provide LLMs with a precise set of tools tailored to a specific task. The authors propose a three-stage approach called Precision-driven Tool Recommendation (PTR) to address this problem. PTR addresses the limitations of existing tool retrieval methods, which often focus on quantity over quality, leading to inefficiencies in tool selection. The proposed method consists of three stages: Tool Bundle Acquisition, Functional Coverage Mapping, and Multi-view-based Re-ranking. It leverages historical tool usage patterns to identify relevant tools, assesses their functionality against specific queries, and refines the selection through a multi-view ranking process. Additionally, the authors introduce a new dataset, RecTools, and a specialized evaluation metric, TRACC, to effectively measure the performance of tool recommendations. Extensive experiments demonstrate the approach's accuracy and potential to improve LLMs' problem-solving capabilities.

**Strengths:**

* The introduction of the tool recommendation problem, alongside the construction of the RecTools dataset and the TRACC metric, significantly advances research in this area.

* The PTR approach effectively dynamically adjusts the recommended toolset, enhancing both precision and effectiveness. Additionally, the comprehensive experiments and analyses provide strong validation for the proposed approach.

**Weaknesses:**

* My main concern is about the definition of “tool recommendation”. In traditional recommendation situations, different users with the same historical behaviors may prefer different items. However, in the tool recommendation, I think there is only one ground truth for a given query. This is more like “prediction” than “recommendation”.



* The paper does not provide a thorough comparison with existing tool retrieval or recommendation methods beyond the selected baselines. It would be helpful to understand how PTR performs compared to a wider range of state-of-the-art tool retrieval approaches.

* The paper does not discuss the potential computational and memory overhead of the PTR approach, which could be an important consideration in real-world deployments. It is recommended to report token consumption and model complexity for this work, as these factors are essential for understanding the computational demands of the proposed models.

**Questions:**

Please refer to the "Weaknesses".

---

### Official Review · Reviewer_JYRm · 2024-11-02

**Soundness:** 2
**Presentation:** 3
**Contribution:** 2
**Rating:** 3
**Confidence:** 5

**Summary:**

To provide LLM with comprehensive yet non-redundant tool sets, this paper proposes a novel zero-shot tool retrieval approach called Precision-driven Tool Recommendation (PTR). PTR includes three key stages via different prompts: tool bundle acquisition, functional coverage mapping, and multi-view-based reranking. Additionally, the paper introduces a new tool retrieval dataset, RecTools, and a new metric, TRACC. Experiments on three datasets demonstrate the effectiveness of the proposed PTR.

**Strengths:**

1. The paper is well-organized and easy to read.
2. The proposed method is intuitive and experiments on three datasets show its effectiveness in zero-shot tool retrieval.
3. The code is publicly available, enhancing reproducibility.

**Weaknesses:**

1. The design of this work emphasizes that tool retrieval results should be comprehensive and non-redundant but lacks an explanation of the necessity of non-redundancy. For instance, the concept of "oversolving" mentioned in line 233 is not supported by specific quantitative experiments illustrating the impact of oversolving.
2. The definition provided in Definition 1 is redundant. When the recommended tool set $T_{recommend}$ equals the ground truth tool set $T_{ground}$, the cardinality constraint $|T_{recommend}| = |T_{ground}|$ is inherently satisfied.
3. The paper lacks an explanation of the necessity for proposing RecTools. The proposed RecTools dataset is not compared with ToolBench[1], which appears to encompass all the characteristics of RecTools. Additionally, during dataset construction, only GPT-4 was employed to combine queries and tool descriptions to determine whether the provided toolset can achieve an "Exact Solving" outcome for each query, without actually calling the tools to generate responses. This approach may reduce the confidence in the dataset labels.  Moreover, from the main experimental results in Table 2, the metrics on RecTools are significantly higher than those on ToolLens, with NDCG@K approaching 0.9 even under the zero-shot retrieval setting, indicating that this dataset may be less challenging.


[1] Toolllm: Facilitating large language models to master 16000+ real-world apis

**Questions:**

1. The content studied in this work does not differ from previous tool retrieval research, and the formulation is consistent. Why introduce the term "tool recommendation" instead of continuing to use "tool retrieval"? This change seems unnecessary.
2. The work attempts to select comprehensive and non-redundant tools by combining LLMs with a multi-view-based reranking prompt. However, selecting truly suitable tools based solely on semantic analysis of tool descriptions may not be effective, especially when many tool descriptions are "No description available," as seen in the appendix examples. Have the authors considered utilizing past interaction information or feedback from the LLM after actually invoking the tools to improve learning?
3. The proposed metric appears to be positively correlated with Recall@K and NDCG@K. Could you provide more explanation about the characteristics and necessity of this metric?
4. The experiments analyze only the results of tool retrieval without examining the actual impact on the downstream LLM-generated responses. As an intermediate step in LLM tool learning, tool retrieval should be evaluated based on the final results of downstream tasks for a more accurate assessment.

Overall, I am willing to increase my score during the rebuttal phase if the authors can address the concerns mentioned above.

---

### Official Review · Reviewer_feUu · 2024-11-03

**Soundness:** 2
**Presentation:** 2
**Contribution:** 3
**Rating:** 3
**Confidence:** 4

**Summary:**

This paper focuses on the crucial aspect of tool recommendation, addressing the issues of under-selection and over-selection present in existing methods. The authors propose a novel tool recommendation approach to achieve precise recommendations, ensuring completeness without redundancy. Additionally, the authors introduce a new tool recommendation dataset and an effective evaluation metric specifically designed to assess tool recommendation for LLMs. Extensive experiments demonstrate the effectiveness of PTR over existing methods.

**Strengths:**

1. This paper addresses a highly important problem: how to provide LLMs with a toolset that is both complete and non-redundant.
2. The paper introduces a new dataset, RecTools, and an effective evaluation metric, TRACC, specifically designed to assess tool recommendation for LLMs.

**Weaknesses:**

1. Some parts of the paper lack clarity. For example, it does not explain how the vector representation of bundles is obtained, nor does it mention the ordering of tools within bundles when calculating metrics in the experimental section.
2. The paper could strengthen its experimental evaluation by including more baseline methods for comparison, particularly in the tool retrieval domain. Recent works, such as the following [1-3], could provide a more robust comparison:

[1] Enhancing Tool Retrieval with Iterative Feedback from Large Language Models, EMNLP 2024.

[2] Re-Invoke: Tool Invocation Rewriting for Zero-Shot Tool Retrieval, EMNLP 2024.

[3] Towards Completeness-Oriented Tool Retrieval for Large Language Models, CIKM 2024.

3. The paper emphasizes that missing or redundant tools can degrade the problem-solving performance of LLMs; however, it lacks experimental validation for this claim. Additionally, there is no experimental verification showing that improvements in the proposed tool recommendation method lead to enhancements in downstream tool learning performance.

**Questions:**

1. What advantages does the proposed method have compared to a paradigm where task decomposition is performed first, followed by tool retrieval for each subtask?
2. Additionally, in Table 3, the performance seems to improve even without the Tool Bundle Acquisition step, showing better results than using BM25 and Contriever to acquire the tool bundle initially. Could the authors clarify why this is the case?
3. As far as I know, the ToolBench dataset also meets the requirements of the proposed RecTool dataset. Could the authors clarify the differences between RecTool and ToolBench?

---

### Note · Authors · 2024-12-11

I have read and agree with the venue's withdrawal policy on behalf of myself and my co-authors.